# The Effect of Scale Insects on Growth Parameters of cv. Chardonnay and cv. Sauvignon Blanc Grapevines Grown in a Greenhouse

**DOI:** 10.3390/ijms24021544

**Published:** 2023-01-12

**Authors:** Paul D. Cooper, Thy T. Truong, Andras Keszei, Teresa Neeman, Kenneth W. Webster

**Affiliations:** 1Ecology & Evolution, Research School of Biology, The Australian National University, Canberra, ACT 2601, Australia; 2Proteomics and Metabolomics Platform, School of Agriculture, Biomedicine and Environment, La Trobe University, Melbourne, VIC 3086, Australia; 3Biological Data Science Institute, ANU College of Science, The Australian National University, Canberra, ACT 2601, Australia

**Keywords:** *Parthenolecanium*, grapevine pests, *Vitis vinifera* L., grapevine volatiles, genotype specific response, HS-SPME-GC/MS, overcompensation, cultivar differences, salicylic acid (SA) pathway

## Abstract

Plants can respond to insects that feed with stylet mouthparts using various processes that are initiated via the salicylic acid metabolic pathway. In Australia, scale insects of the genus *Parthenolecanium* can cause economic damage to grapevines as they feed on the vines and produce honeydew as a waste by-product, which supports the growth of black sooty mould on fruit and leaves, potentially affecting the plant growth and yield. Using rootlings of Sauvignon Blanc (SB, resistant) and Chardonnay (Char, susceptible), the growth and production of volatile organic compounds (VOCs) following exposure to scale insect infestations were measured under controlled greenhouse conditions. At harvest, the numbers of scale insects per five leaves were higher on plants infested at the start of the study compared with the control plants. Infested SB had increased dry root and shoot mass compared with the SB control, which was also the case with Char (control and infested). Leaf volatiles differed between cultivars in response to scale infestation. Benzyl alcohol decreased among infested SB plants compared with the other treatments. A change in the salicylic acid pathway as indicated by the change in benzyl alcohol may cause the increased growth in SB associated with the increased scale insect infestation.

## 1. Introduction

Insects can affect plant growth and development as a result of direct feeding, indirect spread of virus or bacteria, or by inducing changes in the plant biochemistry [1,2]. Insects with piercing or sucking mouthparts bypass the direct breakdown of plant material but instead withdraw plant materials by ingesting liquids, either by external digestion of cell contents or by feeding on phloem or xylem [3]. These insects may be able to severely affect plant growth or development, either by removing resources or by introducing viruses [2]. Insects that feed by sucking or piercing have been suggested to induce direct plant defences via the salicylic acid pathway [4,5,6,7], but several indirect defences may also be initiated, such as changes in plant hormones or by attracting beneficial insects with the release of volatile chemicals [4,6,7]. The release of plant hormones can change how the plant grows when sucking insects are present, while the attraction of beneficial insects will reduce the effect of sucking insects as beneficial insects either eat or parasitise the insects feeding on the plant [1,8].

Relatively few studies have examined the VOCs in grapevines in response to insect infestation, with an exception showing changes in vines infested with phylloxera (*Daktulosphaira vitis* Fitch) [9]. Several compounds, including methyl salicylate and benzaldehyde, were increased in infested vines of the cultivar Teleki 5C (*V. berlandieri* × *V. riparia*). Many of the salicylic pathway compounds were known to be elevated in plants attacked by various Hemiptera and even under abiotic conditions, suggesting that grapevines have a typical defence response to attack by sucking insects [10].

Vineyards are large plots with grapevines (*Vitis vinifera* L.) separated into cultivars to ensure that harvesting of grape varieties is easy. Although cultivars are the same species, the development of a cultivar has occurred over several generations, with current propagation normally performed by using clones to ensure that the appropriate cultivar is grown for winemaking. However, these cultivars may vary in characteristics that have wide-ranging advantages for certain habitats and their associated pests as agricultural practices have separated some cultivars for thousands of years [11]. Just as phylloxera has been partially controlled by using rootstock of the North American plants (e.g., *V. riparia*), potentially different cultivars have the ability to regulate other pests that attack and cause plant damage, as suggested for *Epiphyas postvittana* Walker, the light brown apple moth [12].

Scale insects of the genus *Parthenolecanium* were introduced into Australia over 90 years ago [13,14]. Two species have subsequently infested grapevines, *P. persicae* (Fabricius) (grapevine scales) and *P. nr. pruinosum* (frosted scales), both with a single generation per year in Australia. Eggs hatch in late spring, develop through summer, overwinter whilst immature and develop into adults in spring, with all adults being parthenogenic females [15,16]. In extremely dense populations, these scales can reduce growth in grapevines [15], but the major problem is the formation of sooty mould as a result of the honeydew that these species produce [15]. Honeydew is rich in sugars [17] and acts as a substrate for sooty mould growth, which can lead to a reduction in photosynthesis when covering leaves. However, more importantly, the mould can also reduce the economic fruit yield as wineries will either reduce the price of grapes with sooty mould or even refuse to accept the grapes [18]. Climate change is associated with both increases in temperature and humidity in viticultural regions of Australia, conditions that may further increase the chances of sooty mould occurring [19]. As the problem of scale insect infestation has only recently been recognised in Australian vineyards, no estimation of total damage or economic impact has yet been produced.

Previous work has demonstrated that not all grapevine cultivars are susceptible to scale infestation. Simbiken et al. [20] demonstrated that both cv. Sauvignon Blanc and cv. Pinot Noir had a reduced number of scales present in a greenhouse experiment compared with cv. Chardonnay. However, these experiments were performed on vines grown under short-day conditions, so both the growth and fruit yield of plants may not have represented the conditions in the field. In addition, if any one of the plants produced volatiles as a plant defence, it would be difficult to understand what any single cultivar may do in defence against scale feeding. Here, we carry out a more detailed examination of two white-grape-producing cultivars, Sauvignon Blanc (SB, resistant to scale insects) and Chardonnay (Char, susceptible to scale insects), under natural light conditions to examine how the two cultivars respond to the presence of scale—specifically, what potential mechanisms are present that could confer defence against scales in cv. Sauvignon Blanc compared to cv. Chardonnay to cope with insect feeding in both plant response and volatile chemical production.

## 2. Results

### 2.1. Scale Number and Plant Phenotypes

Scale insects were present on most plants by the end of the experiment (47/48 plants; 1 Sauvignon Blanc control plant was free of scales), but the plants deliberately infested at the outset of the study had significantly more scales than control plants (F_[1,33]_ = 15.1, *p* < 0.001) (Table 1). No differences were present in the number of scales between the cultivars.

Plants started producing leaf buds within two weeks of planting. By 30 January, the few plants that produced grapes (Table 2) reached veraison, and all grapes were ripened by 28 February. 

Scale insects on the leaves were second instars (Figure 1A), although some plants (18/24 for Chardonnay, 20/24 for Sauvignon Blanc) had adult females present on woody parts of plants (range 2–10 adults) by April (Figure 1B). 

The presence of the adults on plants did not differ significantly between cultivars or as a result of infestation (Fisher exact test, *p* > 0.05).

Plants showed no effects of disease, with green leaves still produced despite the presence of the scales (Figure 2), and no significant difference was present with respect to the number of plants producing grapes (Fisher exact test, *p* > 0.05), although the two cultivars differed in the mass of grapes produced and the grape mass increased with the presence of scales (Cultivar, F_[1,10]_ = 5.57. *p* < 0.05, Infested F_[1,8]_ = 6.68, *p* < 0.05) (Table 2).

Differences between the cultivars were present in shoot mass (F_[1,32]_ = 14.9, *p* < 0.001) but not root mass (F_[1,32]_ = 3.86, *p* = 0.06), but most of the difference was when Sauvignon Blanc had been infested with scale insects (shoot interaction F_[1,33]_ = 6.33, *p* = 0.02, root interaction F_[1,32]_ = 7.01, *p* = 0.01) (Table 3 and Figure 2). No difference was found in the shoot:root ratio between Chardonnay and Sauvignon Blanc. The addition of scale insects increased the mass of both roots and shoots on Sauvignon Blanc, but no increase was observed in Chardonnay. The increases in the mass of roots and shoots in the Sauvignon Blanc plants were proportional, so no significant change in the root:shoot ratio was present with the heavier infestation of scale insects.

### 2.2. Leaf Volatile Chemistry

Eighty-eight volatile compounds were identified from the leaves (Appendix A), although the relative proportions did differ depending on the cultivar and whether scale insects were present. A principal component analysis on normalized peak areas showed that the first principal component (19.7% of variance) separated the cultivars and that there was a slight shift to the left side of the plot when with scale insect infestation when plotted against the second principal component (Figure 3). 

Several compounds were identified that were important for the differences in principal component analysis by variable importance by projection (VIP) scores, with globulol and 3-oxo-α-ionol being the most important in separating the two cultivars, and four others had VIP scores greater than two. Five of the top fifteen chemicals had different responses for scale infestation in the two cultivars, as shown by the difference in sign or magnitude for the ratios of VIP scores for control and scale-infested plants (Figure 4).

The individual chemicals that have significantly changed between scale-infested and control plants relative to all chemicals measured are listed (Table 4). Most chemicals showed a reduction in the volatile amounts following scale infestation, with the exceptions being methyl salicylate, 2-octanal- and 1-hexanol, 2-ethyl- in cv. Chardonnay. Quantification of some of the chemicals for individual leaves of the two cultivars and the two treatments are presented in Appendix A. 

Two compounds, methyl salicylate and benzyl alcohol, are involved in the salicylic acid response by plants in response to sucking insects. Although methyl salicylate did vary with scale treatment, it was not significantly different between scale-infested and un-infested plants (cultivar F_[1,33]_ = 0.31, *p* > 0.05, infested F_[1,33]_ = 0.34, *p* > 0.05) (Figure 5), but benzyl alcohol differed significantly with cultivar (F_[1,29]_ = 17.83, *p* = 0.0002) and within the Sauvignon Blanc plants (cultivar(infested) F_[1,29]_ = 3.92, *p* = 0.04) (Figure 6).

## 3. Discussion

Plants have several ways of coping with insect pests [4,5], but the plant–pathogen interactions are complex and mechanisms are still the subject of research. The advantage of studying grapevines is that within one species, we can determine how the various cultivars respond to insect pests. In this research, we have shown that neither cultivar was resistant to scale insects as this study indicated that they had similar infestation loads (Table 1). We have shown that a difference is present between the cultivars with respect to volatile compounds (Figure 4) and that slight changes in volatile compounds occur in each cultivar when scale insects are present (Figure 5). However, the big difference in plant response is that the increased scale numbers on Sauvignon Blanc were associated with more robust growth (Table 3 and Figure 2).

The lack of resistance to scale insects shown in cv. Sauvignon Blanc was unexpected, as previous work suggested that the cultivar was resistant [20]. However, the previous experiment was performed in short-day conditions to prevent fruit production, and the allocation of resources for defence could be different under those conditions. In this study, we permitted fruiting to occur by using natural lighting as would occur in the field. The response under natural light suggested that cv. Sauvignon Blanc overcompensated for the presence of scale insects by increasing vegetative growth, although the mass of grapes also increased with scale insects. Previously, an 8% increase in dry mass was measured in Sauvignon Blanc grown under short-day conditions when infested with scale insects [20], also suggesting that this cultivar was overcompensating by growing more when infested. Overcompensation has been reported for several plant species when insects are feeding on the plant [21], and it does not appear to be unusual for plants to increase their growth in response to insect infestation. However, the mechanism behind the increased growth in cv. Sauvignon Blanc is currently unknown, although it would appear to be some form of hormonal change induced by the feeding of the insects, potentially ethylene as it has been reported to be released by other sucking herbivores [22,23]. As the grape mass also increased, slight stress may also be involved in stimulating fruit development, although it may just be another form of overcompensation [21]. Overall, the shoot:root ratio is relatively low for these plants, potentially a result of the plants growing in the polycarbonate greenhouse with the shade cloth, as polycarbonate reduces UV-A transmittance [24], but that might not affect all grapevines [25]. The repressed growth of the other cultivar groups may have made the increased growth of the infected cv. Sauvignon Blanc more obvious as a result.

No change in development time is observed for the majority of scale insects on both cultivars compared with field studies [16,26,27], as second instars would be expected entering the winter period for frosted scale insects (*P. nr pruinosum*). Presumably, the development pattern for the scale insects is independent of the growth pattern of the plants, with the timing of moulting only responsive to temperature and light conditions. The limited temperature range within the greenhouse would have influenced the plant growth as well. Climate change and an increase in temperature could increase both plant and insect development, but to what extent this could lead to a complete second generation of scale insects on grapevines is still unknown. The appearance of some adult females on the plants did suggest that individual insects could develop and moult into the third instar and then become adult females under greenhouse conditions, but the numbers of adults were low and we harvested the plants just as egg development would have been initiated.

We did find scale insects on nearly all plants at harvest, whether infested in December or used as so-called scale-free control plants. Clearly, the enclosed environment of a greenhouse can permit the transfer of scales following hatching as found previously [20]. This mode of infection is probably replicated in the field as scale insects may be distributed by air movements within the vineyards [28] (young and females do not have wings), although adjacent vines may also be infested by nymphs moving along the cordons and wires from one plant to another. However, as far as we know, no study has determined how fast scale insects move in the field for comparison with this work, although scale insects appear to be becoming a greater problem in Australian vineyards [18,19]. It is likely that as vines age, the plant response changes to scale insect infestation [29].

The volatile compounds differentiate the two cultivars, with only minor changes observed with the presence of scale insects. Methyl salicylate, which is known to be involved in the plant defence against sap-sucking insects [5], such as scale insects, did not differ significantly between the cultivars, with only slight changes in the infested cultivars (Figure 6). Benzyl alcohol, which is also part of the salicylic acid biochemistry pathway [30], did decrease among scale-infested cv. Sauvignon Blanc, suggesting that there may be changes in the allocation of resources within the salicylic acid response and that the change in allocation may affect the benzyl alcohol content in plant leaves. Other volatiles also varied between scale-infested cv. Sauvignon Blanc and cv. Chardonnay (Table 4 and Figure 5), but we do not know whether these differences influence the change in growth of the cultivars observed or whether these changes are something other than a cultivar-specific response to herbivory. The volatile compounds differentiate the two cultivars, with only minor changes observed with the presence of scale insects. Globulol and 3-oxo-α-ionol are indicated as having the greatest effect on the separation of the cultivars (Figure 5), but their effect does not appear to differ when scale insects are present. Globulol, a terpene, has not been reported in previous studies of volatiles in *V. vinifera*, but 3-oxo-α-ionol has been found previously [31,32]. Globulol has been reported from *Eucalyptus* sp. and has been reported to have insecticidal activity against mosquito larvae [33]. The next three compounds in importance by VIP scores (phenol, 2-methoxy-3-(2-propenyl)-, 1-nonanol and benzyl alcohol) either changed the sign of the ratio between the two cultivars (first two) or had a much greater absolute value between the two cultivars (benzyl alcohol). These responses suggest that the presence of scale insects influenced the volatile chemical response as indicated by the slight change in the principal component analysis when scale insects were present. The first compound phenol, 2-methoxy-3-(2-propenyl)- has been reported to increase the mortality of grain weevils (*Sitophilus oryzae* (L.) and *Oryzaephilus surinamensis* (L.)) [34], while 1-nonalol has been reported to be involved in pheromones of beetles [35,36]. Also of interest is that trans-β-ionone has been reported as a beetle attractant [37], suggesting that changes in both 3-Oxo-α-ionol and 1-nonanol may improve ladybird beetle recruitment to scale-infested plants. Plants that are fed on by aphids have been shown to increase the volatile release of compounds that are attractive to beneficial insects, such as ladybird beetles [8]. Fieldwork has shown that some cultivars, such as Pinot Noir, have a higher incidence of both ladybird beetles and parasitic wasps (*Metaphycus* sp.) [26], but in the greenhouse, both of those insects are excluded. As this may be the most common pathway for volatile compounds to aid plant defence, we cannot exclude that the observed differences in VOCs could be defences under natural conditions.

The role of VOCs in grapevine defence against insects needs to be examined across a range of cultivars to determine to what extent similarities are present in their use as direct or indirect control processes. Principal component analysis using chemicals extracted from leaves collected from several cultivars in the field suggests that a difference exists between resistant and susceptible cultivars [19], but whether this represents a response to scale insects or is simply correlated with an apparent resistance difference requires further study. Riesling cultivars showed very little change in gene expression following relatively short exposures (6 and 96 h) to the mealybug *Planococcus ficus* (Signoret) [38], while other cultivars may have some tolerance or resistance to mealybugs [39]. Lawo, Weingart, Schuhmacher and Forneck [9] indicate that volatiles varied when phylloxera were present on a single resistant cultivar, but the time over which this change occurred was relatively short. Our work suggests that a change in metabolic pathways in grapevines occurs when scale insects are present but that the response may take longer compared with the response to phylloxera. However, the cultivars may have different responses as cv. Riesling may produce volatiles representative of susceptible cultivars, such as cv. Chardonnay, and therefore may have reduced defensive responses [19]. In contrast, resistant cultivars would have a different response pattern of volatiles, and differences may be partially dependent upon whether the plant has red or white grapes as more red cultivars may be resistant than white cultivars [16]. Further work is examining how resistant red cultivars may respond to scale insect infestation.

Scale insects are causing difficulties in vineyards by increasing the number of vines with sooty mould. As the number of eggs produced per female is between 300 and 500 per individual [16], infestations can quickly become serious, leading to economic damage. Our study used a limited number of females to infest each plant, but in the field, the number of females present on each plant is usually much greater (>25) [29]. Further work is necessary to determine how to prevent scale insects from becoming a serious recurring economic pest in vineyards.

## 4. Materials and Methods

Rootlings of Chardonnay (Clone 95) and Sauvignon Blanc (Clone F4V6) (24 of each) were obtained (Glenavon Nursery, Langhorne Creek, South Australia) and planted in pots (volume 4.5 L) containing soil mixture (Martins mix) and slow-release fertilizer (Osmocote^®^) on 25 September. The plants were grown on two benches in a temperature-controlled (evaporative coolers) polycarbonate greenhouse (25 °C day:15 °C night) for 9 weeks. A 70% shade cloth covered the greenhouse starting in the first week of December to limit the sunlight and ultraviolet wavelengths during the Austral summer months.

Plants were organized in a row–column design (Figure 7) that was generated using CycDesigN (VSN International, Hemel Hempstead, England UK) to account for position effects (row, column, bench) on the two experimental factors of interest: cultivar (2 levels) and scale (2 levels).

Plants were grown together for nine weeks and then organized into the row–column pattern before infesting with scales. Plants were trimmed on 8 November and again 1 week prior to the introduction of the scale insects (lateral shoots only to limit plant–plant contact [40]). The cuttings were dried overnight at 50 °C and weighed (±0.1 g, Sartorius PT1500) to measure the shoot growth for each plant without scales. No effort was made to control either powdery mildew or sooty mould, both of which were observed on the leaves of only a few vines during the trial. We did not treat the plants for those diseases in case any exogenous chemicals used could affect the secondary metabolites produced by plants in response to the scale insects.

Frosted scale insects (*Parthenolecanium nr*. *pruinosum*) were collected from grapevines (cv. Shiraz) in the Mt. Majura vineyard (35.23° S, 149.19° E) in the first week of December, returned to the laboratory and determined to be gravid (containing eggs or first-instar scale insects) under a dissecting microscope (Leica MZ8). Five adult frosted scale insects were used to infest the plants as designated in the design, twelve plants for each cultivar, using the cotton technique [20], with the cotton attached at the first branching region (Figure 2). Scales were allowed to hatch and distribute within the plant, with periodic observations to determine scale infestation. The greenhouse temperature was controlled with evaporative cooling and fans (25 °C day, 15 °C night), so although the plants were arranged to avoid direct movement of first-instar scale insects between infested and control plants, the air currents generated by the fan could have distributed the first instars throughout the greenhouse as scales are known to distribute using wind [28]. No ant control was undertaken, so ants were present on plants and attended the scale insects, minimizing honeydew on plants and preventing sooty mould. No scale insect movement by ants carrying insects was ever observed.

Plants were allowed to grow until early April when all plants were harvested over 2 weeks. Grape bunches were removed with secateurs, the number of bunches recorded and then weighed (A & D 200 HL balance, ±0.1g). Five leaves per plant within 15 cm where scales were introduced were removed at the time of harvest and counts of the number of scales present were recorded with the aid of a dissecting microscope (Zeiss SMXX) as an indicator of scale infestation. As no difference in leaf area between these cultivars was found previously [20] and leaves of a similar size were selected, the leaf area examined was assumed to be similar. The roots of each plant were washed and the roots and shoots were separated and dried at 50 °C for 24 h as carried out previously [20]. The dry mass (±0.1 g) of each root and shoot was measured with a balance (Sartorius PT1500).

Prior to harvesting the plants, five leaves of each plant that were at least 35 cm from the region where scales were introduced were removed using secateurs. The leaves were placed in a labelled paper bag without handling. The secateurs were cleaned with 80% ethanol between each plant to ensure that any adhering plant material was removed. Immediately after placing leaves in the labelled bag, the bag with leaves was inserted into a dry shipper (pre-treated with liquid nitrogen) to immediately freeze both the plant and bag. The plants and bag were then placed into a freezer (−80 °C). Plants and bags were freeze-dried (Virtis^TM^ SP Scientific, Ipswich, UK). Dried leaf pieces (0.5 g) of nine plants for each cultivar and treatment (total 36 plants) were placed in 20 mL headspace (HS) vials (Agilent Technologies, Mulgrave, Victoria, Australia) and capped for subsequent volatile analysis, along with QC laboratory blanks (empty HS vials exposed to ambient laboratory conditions during standard and sample preparation before capping) and 10 µg/L of QC standard mix (60 VOCs) and 10 µg/L of n-hydrocarbon standard mix (C_9_–C_22_). Following the optimised HS-SPME-GC/MS methodology from Rivers, et al. [41], the VOCs of the leaves, QC laboratory blanks, and QC standards were randomised and subsequently incubated (70 °C for 5 min), extracted and adsorbed (40 min) onto a general-purpose bipolar 50/30 µm PDMS/DVB/CAR SPME fibre (Supelco, Sigma-Aldrich, Castle Hill, Australia) using an MPS 2 Gerstel Multipurpose Sampler headspace solid-phase microextractor (HS-SPME, Gerstel GmbH & Co. KG, Mülheim an der Ruhr, Germany), desorbed in the GC inlet (10 min at 250 °C), and measured by gas chromatography–mass spectrometry (GC/MS; Agilent Technologies, Palo Alto, CA, USA) using a 30 m × 0.25 mm × 0.25 μm non-polar Agilent J&W VF-5 ms column with a 10 m EX-Guard column. The temperature programme was 40 °C for 2 min, increased at 5 °C/min to 150 °C and held for 2 min, then 15 °C/min to 320 °C for 1 min. Ultra-high-purity helium (BOC Australia, Fyshwick, Australia) was used as the carrier gas at a flow rate of 1 mL/min. The temperatures for the quadrupole, ion source, and Aux transferline were 150 °C, 250 °C, and 320 °C, respectively. The splitless mode acquired full-scan MS data (*m/z* 40–500), which was analysed using Agilent Masshunter data analysis software (version B.70). Mass spectra (≥70% confidence) and Kovats non-isothermal RIs were compared with QC standards (including methyl salicylate and benzoic acid), the NIST/EPA/NIH Mass Spectral Library (version 2017), PubChem, and Adams Essential Oils databases for mass spectral and retention index (RI) matching, respectively, to meet the metabolite identification confidence levels set by the metabolomics standards initiatives (MSI) [42,43].

We compared the root and shoot dry masses, shoot:root ratio, and number of scales per five leaves to determine how the infestation or cultivar impacted each parameter. Statistical analyses were performed using R version 3.5.3. Primary statistics were gathered using linear mixed models with the cultivar and infection as fixed factors and bench, row, and column as random factors. For root and shoot analysis, the dry mass of the cuttings prior to scale introduction was included as a covariate. If the significance (*p* < 0.05) was indicated by ANOVA, then pairwise comparisons were made to determine which parameters of the model were significant. Means and standard errors were presented for each parameter measured. For the HS-SPME-GC/MS analysis, we used Metaboanalyst v.4.0 to determine principal component analysis (PCA) and partial least squares discriminant analysis (PLS-DA) on normalised data based on peak areas of the VOCs detected. The PLS-DA was used to determine which volatiles contributed the most to explaining the variation as well as which volatiles varied between cultivars and treatments with variable importance in projection (VIP) scores used to determine which volatiles had the greatest influence on the PLS-DA. Analyses of methyl salicylate and benzyl alcohol were performed using linear mixed models, but with GC/MS batch runs to account for any systemic fluctuations during SPME-GC/MS sequence queue, used as the random factors (1, 2, and 3), as each batch run included only 3 of each cultivar and treatment for only 12 plants in each run.

## Figures and Tables

**Figure 1 ijms-24-01544-f001:**
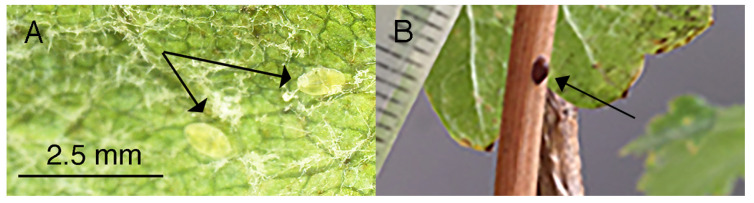
(**A**). Second instars on leaves of cv. Chardonnay. Arrows indicate position of the instars. (**B**). Adult female (arrow) located on woody branch of cv. Chardonnay.

**Figure 2 ijms-24-01544-f002:**
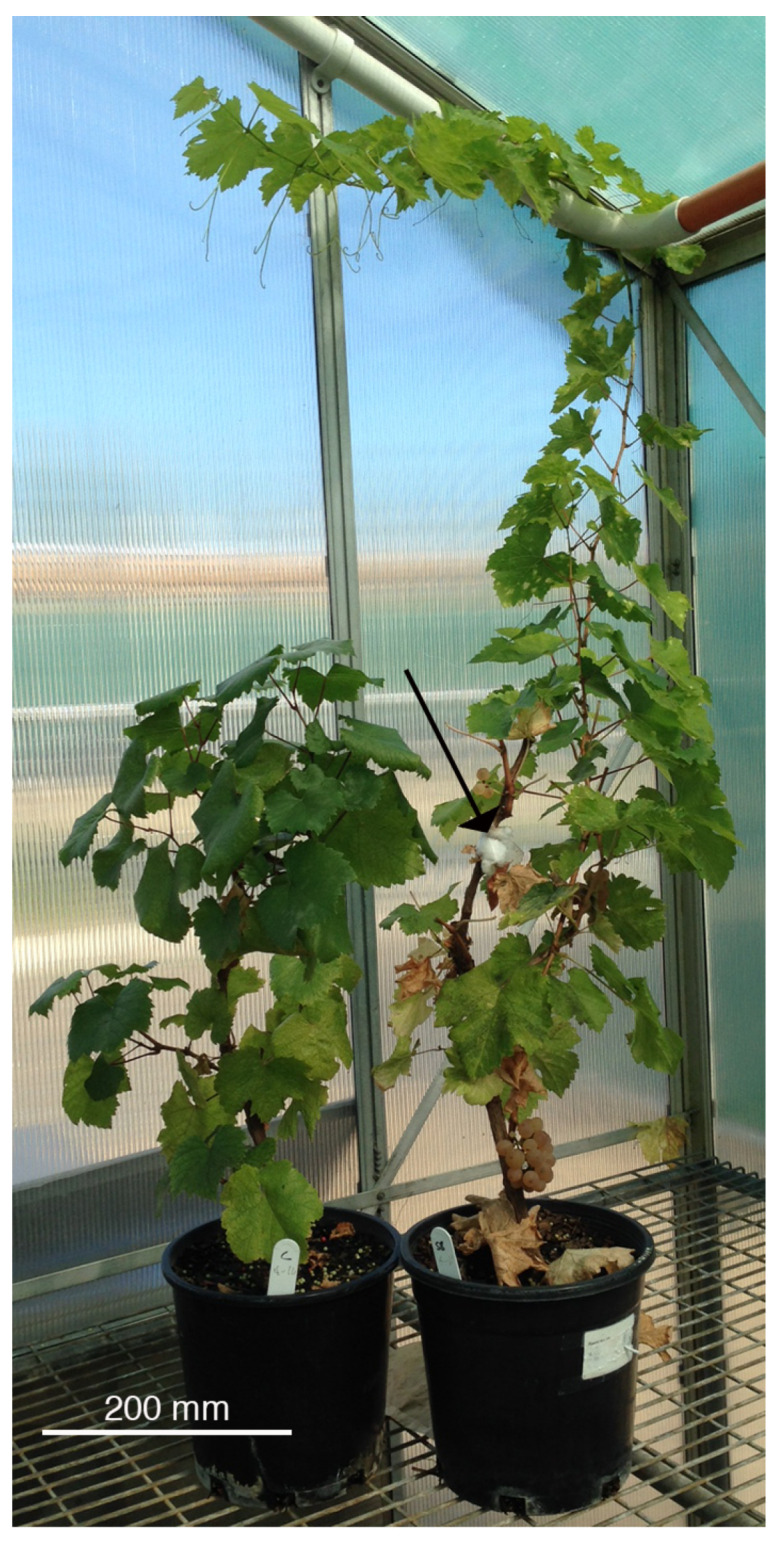
Comparison of Chardonnay (left) and Sauvignon Blanc grown in the greenhouse, with the shoot of Sauvignon Blanc (cotton (arrow) indicates that this was infested with scale insects) being longer than that of Chardonnay.

**Figure 3 ijms-24-01544-f003:**
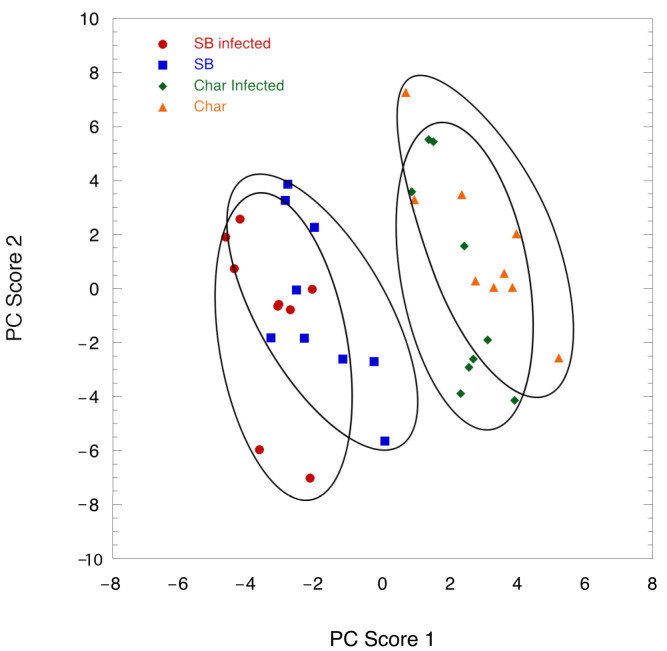
Principal component graph of the cultivars with and without scale infestation. Cultivars are separated by principal component 1, but infestation only shows partial differentiation along both PC scores.

**Figure 4 ijms-24-01544-f004:**
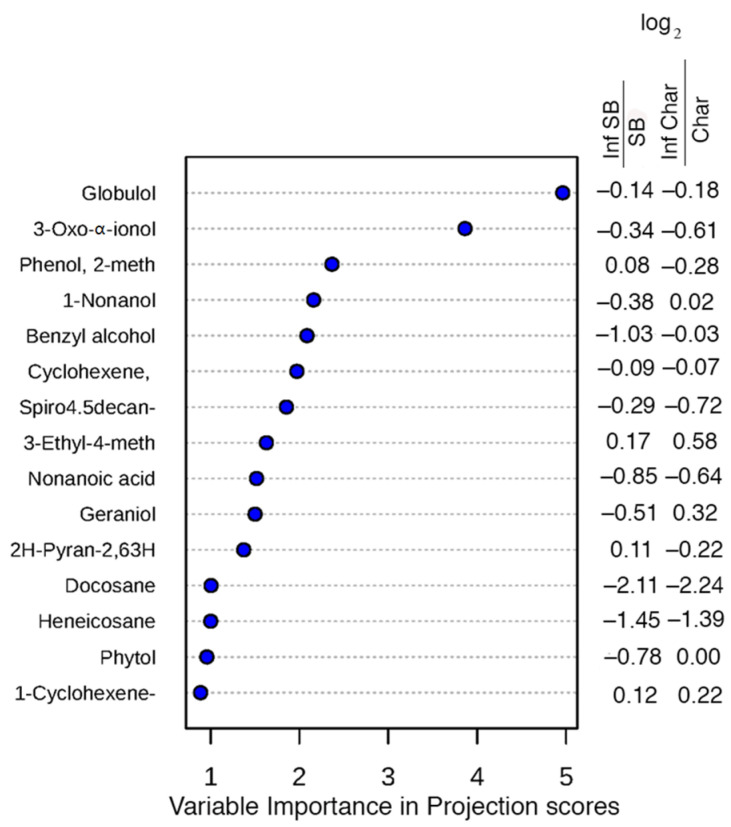
Compounds that have the greatest influence as indicated by principal component scores. Globulol and 3-oxo-α-ionol have the greatest influence and show decreases in vector score in both cultivars with infestation by scale insects. Ratios of scores that changed signs or have large absolute changes indicate compounds that change with scale infestation.

**Figure 5 ijms-24-01544-f005:**
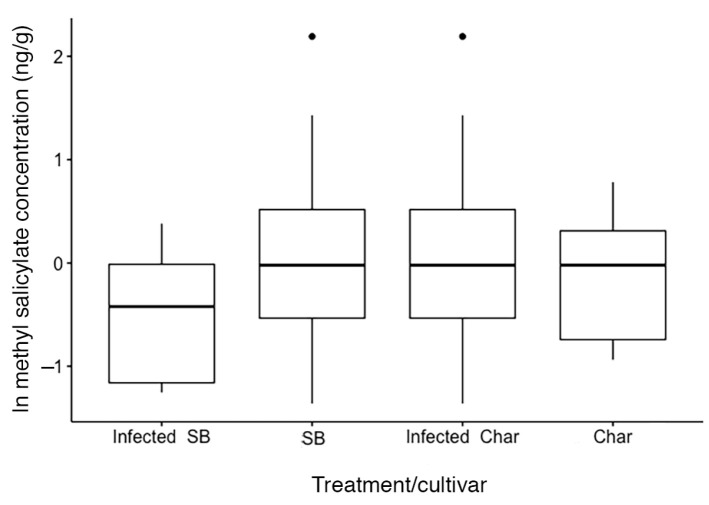
Box plots of methyl salicylate showing that infested Sauvignon Blanc had lower mean concentrations but that overall this compound did not significantly differ either between cultivars or between treatments across the two cultivars.

**Figure 6 ijms-24-01544-f006:**
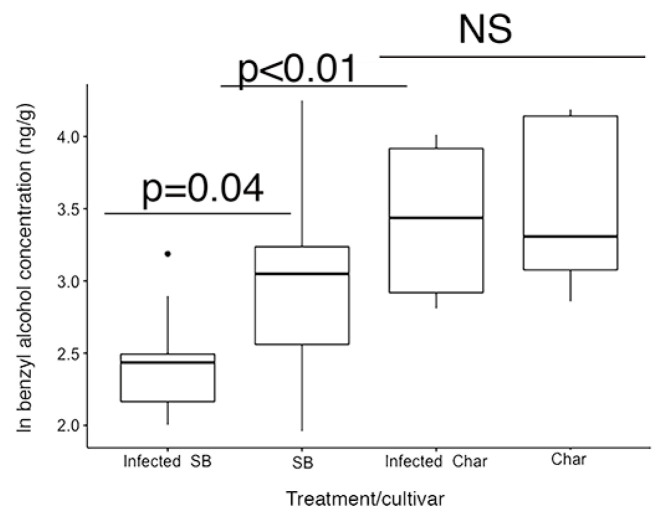
Benzyl alcohol decreases significantly in Sauvignon Blanc (SB) cultivar when infested with scale insects, while no difference was observed between the infested and control Chardonnay (Char) cultivars. NS = not significant.

**Figure 7 ijms-24-01544-f007:**
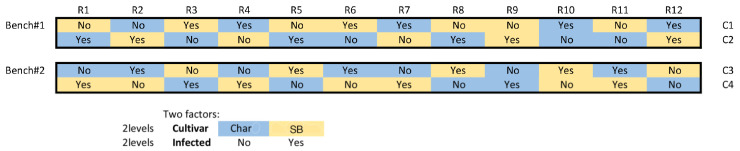
Design of experiment: row–column design (CycDesigN) with 12 plants of each cultivar as control and as infested with scales and balanced between rows and columns on two benches in the greenhouse. The design (cultivar * infected) with random effects of bench (1–2), column (C1–4), and row (R1–12) was used in the linear model analysis using R.

**Table 1 ijms-24-01544-t001:** Mean number of scales on five leaves removed from the Chardonnay and Sauvignon Blanc cultivars in relation to whether the plants were control or initially infested plants. Mean ± S.E. n = 12 for each treatment. The same letters indicate no significance, while different letters indicate significant difference (*p* < 0.001).

	Chardonnay	Sauvignon Blanc
Control	60.6 ± 23.7 ^a^	31.2 ± 23.7 ^a^
Infested	143.4 ± 23.7 ^b^	114.0 ± 23.7 ^b^

**Table 2 ijms-24-01544-t002:** Number of plants that produced grapes and mass of grapes produced for each cultivar and treatment. Mean mass of grapes is determined only for plants that produced grapes. Mean ± S.E. No difference was found in the number of plants producing grapes in any category (Fisher’s exact test), but Chardonnay produced a greater mass of grapes than Sauvignon Blanc and infested plants of both cultivars increased grape production (*p* < 0.05). Different superscript letters indicate significant differences (*p* < 0.05).

	Chardonnay		Sauvignon Blanc	
	Number with Grapes	Mean Mass of Grapes (g)	Number with Grapes	Mean Mass of Grapes (g)
Control	5	67 ± 15.6 ^a^	3	23 ± 24.7 ^b^
Infested	1	112 ± 32.7 ^c^	6	68 ± 14.5 ^d^

**Table 3 ijms-24-01544-t003:** Comparison of root and shoot mass and shoot:root ratio for cultivars of Chardonnay and Sauvignon Blanc, with or without scale infestation at beginning of experiment. Mean ± S.E. Different letters indicate statistical differences (*p* < 0.01) among plant parts either between cultivars or with infestation with scales.

	Chardonnay			Sauvignon Blanc		
	Root (g)	Shoot (g)	Shoot:Root	Root (g)	Shoot (g)	Shoot:Root
Control	170 ± 32.4 ^a^	39 ± 2.8 ^c^	0.3 ± 0.06 ^f^	160 ± 32.4 ^a^	42 ± 2.8 ^c^	0.3 ± 0.06 ^f^
Infested	151 ± 32.2 ^a^	35 ± 2.8 ^c^	0.3 ± 0.06 ^f^	216 ± 32.2 ^b^	51 ± 2.8 ^d^	0.2 ± 0.06 ^f^

**Table 4 ijms-24-01544-t004:** Volatile organic compounds that varied with the infestation of scale insects. Data are ratios of compounds from infested and control plants and the log_2_ of those ratios. These changes are significantly different from the overall chemical changes. A negative value indicates that chemicals in infested plants are reduced compared to control plants.

Cultivar/Volatile	Fold Change	log_2_ Fold Change
Sauvignon Blanc		
Docosane	0.233	−2.10
Heneicosane	0.365	−1.45
Methyl salicylate	0.369	−1.44
Tricosane	0.416	−1.27
Benzyl alcohol	0.490	−1.03
Nonanoic acid	0.554	−0.85
Phytol	0.584	−0.78
Chardonnay		
Docosane	0.218	−2.20
Tricosane	0.319	−1.65
Heneicosane	0.381	−1.39
Eicosane	0.594	−0.75
Spiro4.5decan-7-one, 1,8-dimethyl-8,9-epoxy-4-isopropyl-	0.606	−0.72
Nonanoic acid	0.642	−0.64
Methyl salicylate	1.527	0.61
3-Oxo-α-ionol	0.655	−0.61
2,5-Furandione, 3,4-dimethyl-	0.663	−0.59
2-Octenal, E-	1.524	0.61
1-Hexanol, 2-ethyl-	1.501	0.59

## Data Availability

Data for this study are available through the DataCommons (https://datacommons.anu.edu.au/DataCommons/rest/display/anudc:6170?layout=def:display, accessed on 1 December 2022) of The Australian National University and are available upon request.

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
