# Peer review of "The Effect of Scale Insects on Growth Parameters of cv. Chardonnay and cv. Sauvignon Blanc Grapevines Grown in a Greenhouse"

_ijms, 2023, doi:10.3390/ijms24021544_

Round 1

Reviewer 1 Report

The topic is very interesting. I think the manuscript will be suitable for publication after minor revision. Authors need to add the author's citation to a scientific name for the first time in the text. Where they found significant differences in statistical analysis (p=0.04, p=0.006, etc.), they may write as p<0.05, p<0.01, p<0.001. My comments are given in the attached PDF file and also given below:

Line 46: add author citation to a scientific name.

Line 50: add author citation.

Line 59: add author citation.

Line 91: p=0.0005---- author may write as p <0.001

Figure 1: It will be better if the scale bar is placed at the bottom corner of the image. I also suggest placing the two images (A & B) side by side. Figure 1B need to zoom to increase the visibility of the adult insect on the stem.

Line 104: p>0.05--- mention the value of p.

Line 110: mention the value of p.

Line 112: Cultivar- Is statistical analysis performed between two cultivars? Infested- is the statistical analysis between non-infested and infested?

p=0.04, p=0.03---  author may write as p<0.05

Figure 2. Placing the scale bar at the bottom corner of the image will be better.

Line 122: p<0.001

Line 124,125: p<0.05

Line 167: add p values.

Line 245: Eucalyptus spp.

Line 309: 149.19---- mention direction.

Line 315: 25 degrees, 15 degrees --- Celsius? add.

References--- maintain proper style. Avoid commas after the Last author's name, Use full stops in abbreviated journal names, and bold the year of publication.

Author Response

Line 46: add author citation to a scientific name.

Done

Line 50: add author citation.

Done

Line 59: add author citation.

Done

Line 91: p=0.0005---- author may write as p <0.001

Changed to p<0.001

Figure 1: It will be better if the scale bar is placed at the bottom corner of the image. I also suggest placing the two images (A & B) side by side. Figure 1B need to zoom to increase the visibility of the adult insect on the stem.

Moved scale on 1A and put arrow pointing to adult female insect on B

Line 104: p>0.05--- mention the value of p.

Nothing done as p>0.05, does reviewer mean give value of exact test?

Line 110: mention the value of p.

Nothing done as p>0.05, does reviewer mean give value of exact test?

Line 112: Cultivar- Is statistical analysis performed between two cultivars? Infested- is the statistical analysis between non-infested and infested?

Model is comparing within group Cultivar (Chardonnay vs Sauvignon Blanc) and with group Infested (infested vs control)

p=0.04, p=0.03---  author may write as p<0.05

Done

Figure 2. Placing the scale bar at the bottom corner of the image will be better.

Done

Line 122: p<0.001

Dome

Line 124,125: p<0.05

Done

Line 167: add p values.

Done

Line 245: Eucalyptus spp.

Changed

Line 309: 149.19---- mention direction.

Done

Line 315: 25 degrees, 15 degrees --- Celsius? add.

Done

References--- maintain proper style. Avoid commas after the Last author's name, Use full stops in abbreviated journal names, and bold the year of publication.

I have used the IJMS endnote style.

Reviewer 2 Report

Dear authors,

Very interesting and comprehensive study and important topic for viticulture, especially in a warmer climates. Scale insects can be very dangerous if left out of control, they seem to be slow but they can easily cause 'sudden' colapse of plant. 

This study requires minor changes. Please find the attached pdf of the article with some additional comments.

Wish you all the very best!

Kind regards,

Reviewer

Author Response

Responses to Reviewer 2

Pg 1

  1. Unify abbreviations

Done

  1. Control

Done

  1. Keywords

Added some suggested words

  1. Add references

Added

Pg. 2

  1. Add references.

Done

  1. Re-write sentences 50-55.

Done

  1. Economic problem of pest

Currently unknown as no study has been done on the economic losses to scale insects as Wine Australia has not designated scale insects as a problem.  However individual vineyards have been complaining.

  1. Sauvignon Blanc (as resistant to scale insects) Chardonnay (as susceptible to scale insects)

Change made

  1. Control?

Control inserted

Pg3.

  1. Treatment?

Done.

  1. Arrows indicate position on leaves

Changed

  1. Can this sentence be changed to before Figure 1?

Done.

  1. Or “although the two cultivars differed in mass produced”

Changed to this wording.

Pg.5

  1. “Slight shift to left by scales present on leaves”

Added.

  1. Bi-plot?

Decided to stick with PCA because of simplicity

  1. Simplify legend

Done

Pg. 6

1.Simplify information on VIP

Done

Pg.7

  1. Cultivar

Encompasses comparison between two cultivars (SB vs Char)

  1. Something like:

Included in text as reviewer phrased it.

Pg.8.

1.Robust growth

Changed

  1. Add references

Done

  1. Add references

Done

  1. Add reference

Done

  1. Discussion of spread of scale insects

Did my best, but again little known about this outside work in my group.

Pg. 9

Thanks for the interesting comments. Have done my best, but again the research needs some outside support.

Pg.10.

  1. Figure change?

Kept the figure nearly the same as the linear modelling was done using Cultivar (SB vs Char) and Infested (yes or no) and explained the abbreviations in the legend as they were used as random factors (location of plant).

  1. Levels

Hope I have made this clearer as it was 2 treatments x 2 cultivars

  1. Location of scale introduction

Yes, “in the first branching point” added to text.
